# Differential Gene Expression in Porcine Lung Compartments after Experimental Infection with *Mycoplasma hyopneumoniae*

**DOI:** 10.3390/ani14091290

**Published:** 2024-04-25

**Authors:** Rubén S. Rosales, David Risco, Obdulio García-Nicolás, Francisco J. Pallarés, Ana S. Ramírez, José B. Poveda, Robin A. J. Nicholas, Francisco J. Salguero

**Affiliations:** 1Instituto Universitario de Sanidad Animal y Seguridad Alimentaria (IUSA), Veterinary Faculty, University of Las Palmas de Gran Canaria, Trasmontaña s/n, 35416 Arucas, Spain; ruben.rosales@ulpgc.es (R.S.R.); anasofia.ramirez@ulpgc.es (A.S.R.); jose.poveda@ulpgc.es (J.B.P.); 2Unidad de Histología y Anatomía Patológica, Departamento de Medicina Animal, Veterinary Faculty, University of Extremadura, Avenida de la Universidad, s/n, 10003 Cáceres, Spain; 3Institute of Virology and Immunology (IVI), Sensemattstrasse 293, 3147 Mittelhäusern, Switzerland; obdulio.garcia-nicolas@ivi.admin.ch; 4Department of Infectious Diseases and Pathobiology, Vetsuisse Faculty, University of Bern, 3012 Bern, Switzerland; 5Pathology and Immunology Group (UCO-PIG), Department of Anatomy and Comparative Pathology and Toxicology, UIC Zoonosis y Enfermedades Emergentes ENZOEM, University of Córdoba, International Excellence Agrifood Campus “CeiA3”, 14001 Córdoba, Spain; fpallares@uco.es; 6The Oaks, Nutshell Lane, Farnham, Surrey GU9 0HG, UK; robin.a.j.nicholas@gmail.com; 7School of Veterinary Medicine, University of Surrey, Daphne Jackson Rd, Guildford GU2 7AL, UK; m22119@surrey.ac.uk

**Keywords:** *Mycoplasma hyopneumoniae*, laser-capture microdissection, gene expression, IL-8, IL-10, IFN-α

## Abstract

**Simple Summary:**

*Mycoplasma hyopneumoniae* (Mhyo) is a major porcine pathogen worldwide. Understanding its interaction with the pig immune system is crucial for effective disease control. This study evaluates the use of minimal tissue samples from different lung compartments to analyze this interaction in detail, in addition to the microscopic and macroscopic evaluation of lung lesions. Key findings reveal strain-specific virulence variability and a differential cytokine expression in the lung compartments tested, highlighting the relevance of Th1 and Th2, and a potential role for Th17-mediated immune responses in Mhyo infection. The data analyzed shed light on the complex nature of Mhyo infection and its interplay with the pig’s immune system, potentially aiding in the development of better disease management strategies.

**Abstract:**

*Mycoplasma hyopneumoniae* (Mhyo) is the causative agent of porcine enzootic pneumonia (EP), as well as one of the main pathogens involved in the porcine respiratory disease complex. The host–pathogen interaction between Mhyo and infected pigs is complex and not completely understood; however, improving the understanding of these intricacies is essential for the development of effective control strategies of EP. In order to improve our knowledge about this interaction, laser-capture microdissection was used to collect bronchi, bronchi-associated lymphoid tissue, and lung parenchyma from animals infected with different strains of Mhyo, and mRNA expression levels of different molecules involved in Mhyo infection (ICAM1, IL-8, IL-10, IL-23, IFN-α, IFN-γ, TGF-β, and TNF-α) were analyzed by qPCR. In addition, the quantification of Mhyo load in the different lung compartments and the scoring of macroscopic and microscopic lung lesions were also performed. Strain-associated differences in virulence were observed, as well as the presence of significant differences in expression levels of cytokines among lung compartments. IL-8 and IL-10 presented the highest upregulation, with limited differences between strains and lung compartments. IFN-α was strongly downregulated in BALT, implying a relevant role for this cytokine in the immunomodulation associated with Mhyo infections. IL-23 was also upregulated in all lung compartments, suggesting the potential involvement of a Th17-mediated immune response in Mhyo infections. Our findings highlight the relevance of Th1 and Th2 immune response in cases of EP, shedding light on the gene expression levels of key cytokines in the lung of pigs at a microscopic level.

## 1. Introduction

*Mycoplasma hyopneumoniae* (Mhyo) is the causative agent of porcine enzootic pneumonia (EP), as well as one of the main bacterial pathogens involved in the porcine respiratory disease complex (PRDC) [1]. It has major economical and animal health implications for the pig industry worldwide, linked to diminished feed conversion rates and increased management costs, due to antimicrobial treatments, vaccination, and mortality associated with secondary infections [2].

The first step in the complex host–pathogen interaction between Mhyo and the porcine respiratory tract is represented by the adhesion of this microorganism to the ciliated epithelial cells of the trachea, bronchi, and bronchioles. By doing so, Mhyo is able to overcome the mucociliary clearance mechanism, one of the first protection barriers to infection within the respiratory tract. This fact, in combination with Mhyo’s ability to induce ciliostasis, loss of respiratory cilia, and epithelial death cells, in combination with an increase in mucin secretion by the respiratory epithelial cells, facilitates the colonization of the lower respiratory tract by Mhyo and other secondary/opportunistic pathogens [3,4,5,6]. The Mhyo adhesion process is intricate, involving an ample set of adhesins, led by P97 and P102 and their paralogs, although more than 35 Mhyo surface proteins have been linked to cell adhesion [5]. In addition, some portions of these adhesion proteins have the ability to bind plasminogen, which, after cleavage, is transformed into plasmin, a molecule that allows the microorganism to weaken the host cell junctions and modulate cell signaling, including the upregulation of proinflammatory cytokine expression and stimulation of chemiotaxis [7]. Additionally, Mhyo lipoproteins found on the surface of this pathogen have been associated with the induction of apoptosis of host epithelial and immune cells [8], playing a central role in immunosuppression associated with EP [5,9].

As well as the damage to ciliated epithelial cells, the pathological changes induced in the porcine lung by Mhyo include the initial accumulation of neutrophils and macrophages within the airway lumina [10]. The microscopical lung lesions progress with the development of peribronchiolar/bronchial and perivascular lymphocytic infiltrates, in combination with the presence of bronchoalveolar exudate and the characteristic proliferation of inducible bronchus-associated lymphoid tissue (BALT) [11].

Immunomodulation caused by Mhyo, and the host response, are essential factors in the progression of EP and in the development of pulmonary lesions [12]. The production of proinflammatory cytokines, such as interleukin (IL) 1β, IL-2, IL-6, IL-8, IL-12, IL-18, interferon-γ (IFN- γ), and tumor necrosis factor-α (TNF-α), at the lesion site has been extensively described [13,14,15,16]. These cytokines are directly involved in the initial stages of Mhyo infection, facilitating the inflammation of the lung and the proliferation of BALT [17,18]. Although inflammatory changes are highly relevant in the initial steps of Mhyo infection, EP is usually defined as a chronic disease [1]. To achieve this chronicity of the infectious process, the pathogen is able to induce an anti-inflammatory response in the host. For example, Mhyo is able to induce the expression of IL-10, a cytokine associated with the inhibition of the phagocytic activity of lymphocytes [15,19]. This immunomodulatory mechanism, in combination with its ability to induce the apoptosis of immune cells, has been described as the most relevant strategy to evade the immune system of the host as it can perpetuate the infectious process for up to 7 months [5,20].

Another aspect of Mhyo escape from the host’s immune system can be found in its ability to invade porcine macrophages and epithelial cells [5,21]. Mhyo has been traditionally described as an extracellular pathogen. However recent in vitro evidence suggests that the invasion of host cells could be a relevant aspect to add the repertoire of strategies used by this microorganism to evade the immune response and to facilitate its dissemination and persistence in the host [22].

Furthermore, differences in Mhyo virulence have been linked to strain-dependent mechanisms, linking the development of more severe lung lesions to strains with a higher capacity to induce cytokine-mediated inflammation and to replicate more efficiently within the respiratory tract [18].

Due to the complexity of host–pathogen interaction between Mhyo and the different porcine lung cells, research efforts have focused on understanding this intricate interplay in cases of EP, which is essential for understanding the pathogenesis of this disease. Methodologies such as laser capture microdissection (LCM) offer the opportunity to analyze individual cell populations and histological structures obtained from histopathology slides, which could allow us to provide a better correlation between histopathological changes and the local immune reaction against Mhyo infections [23]. This technique has been previously described as an effective tool to elucidate the local host response to bacterial and viral infections [24,25], in combination with quantitative real time PCR (qPCR). This offers the opportunity to better understand the pathological mechanisms occurring in the different lung compartments by analyzing the expression levels of various immune response mediators in selected cell populations, while minimizing background noise of other tissue areas.

The aim of this study was to compare the differential expression of cytokine transcripts, quantified by qPCR, in three lung compartments (bronchi, BALT, and lung parenchyma) of pigs after an experimental infection with four different field strains and a known pathogenic strain of Mhyo. LCM was used to collect cellularly homogenous tissue samples from the different lung compartments of infected and mock-infected animals. In addition, Mhyo load in the different lung compartments, and macroscopic and microscopic scoring of the lungs, were also performed, and compared to the cytokine expression levels, to provide a comprehensive view of the local host changes after Mhyo infection. The role of strain variability in terms of immunogenicity and lung damage was also analyzed in order to improve our knowledge of the host–pathogen interaction in cases of EP.

## 2. Materials and Methods

### 2.1. Bacterial Inoculum

Four Mhyo field strains were used in the study (35p06, 168p09, 252p09, and 20p10). These strains were isolated from clinical lung tissue samples submitted for routine diagnostics at the Mycoplasma Group (Animal and Plant Health Agency (APHA), Weybridge, UK). Mhyo 232 [26] was used as known pathogenic strain, with previous evidence of inducing EP-like lesions after experimental infections in pigs [20]. Mhyo strains were propagated in Friis broth [27], until a concentration of 10^7^ color changing units (ccu)/mL was achieved prior to the experimental infection.

### 2.2. Animals and Experimental Desing

Forty-eight five-week-old male and female Landrace × Large White commercial breed piglets were used in the study. Weight on arrival ranged from 10 to 13 kg. Animals were obtained from a high-health-status commercial farm with a history of Mhyo seronegative results and no vaccination against EP. On arrival, nasal swabs from each animal were obtained, and the presence of Mhyo and other porcine mycoplasmas tested by routine mycoplasma culture methods and denaturing gradient gel electrophoresis [28]. In addition, bacterial and viral testing to ensure the negative carrier status of the piglets was also performed, and the absence porcine circovirus 2 (PCV2), porcine respiratory and reproductive syndrome virus (PRRSv), swine influenza virus (SIV), *Bordetella bronchiseptica*, *Actinobacillus pleuropneumoniae*, *Pasteurella multocida*, and *Glaesserella parasuis* was confirmed by qPCR, RT-qPCR, and microbiological culture methods.

Animals were grouped by weight and randomly distributed into six different groups, based on the Mhyo strain used for the experimental infection (A: 232, B: 35p06, C: 168p09, D: 252p09, and E: 20p10), in addition to a sham-inoculated negative control group (F). Animal housing was performed in individual rooms per each group, with no connection between air spaces. Cross-contamination was avoided by strict biosecurity and animal handling methods. Housing was performed at the APHA Weybridge animal housing facilities (UK). The animal study protocol was approved by the Ethics Committee of the Animal and Plant Health Agency (UK) (Reference: 70/7249-4-001).

### 2.3. Experimental Infection

One week after arrival, animals were infected using a transtracheal inoculation route after anesthesia with isoflurane. Once anesthetized, animals were restrained in the supine position with the head and neck slightly bended dorsally. Then, the trachea was located by tactual exploration and immobilized. A 20G needle coupled with a 10 mL syringe was inserted perpendicular to the trachea. A quantity of 10 mL of 10^7^ ccu/mL Mhyo culture was administered to the appropriate infected group. Group F was inoculated with 10 mL of sterile Friis broth.

After the infection, animals were left to recover individually and then moved to each room. A thorough clinical evaluation was performed daily using a humane endpoint scoring sheet until the end of the experiment.

### 2.4. Serum Sample Collection and Mhyo Specific ELISA

Blood samples were collected from the jugular vein using clotting activation serum tubes (BD Vacutainer^®^, Plymouth, UK) prior to infection (day 0), and at 21 and 28 days post-infection (dpi). After blood collection, serum samples were separated by centrifugation (3500× *g*/5 min) and preserved at −20 °C until analysis.

In order to determine the level of Mhyo-specific antibodies after experimental infection in the different experimental groups, serum samples were tested using a commercial blocking ELISA, following the manufacturer’s instructions (IDEIA™ *Mycoplasma hyopneumoniae* EIA kit, Oxoid, Basingstoke, UK).

### 2.5. Gross Pathology and Microscopic Evaluation of Lung Lesions

Animals were euthanized at 28 dpi. At post-mortem evaluation, lungs were collected from each animal, and a macroscopic lesion scoring protocol was applied to each lung, as previously described [29]. This semiquantitative method is based on a maximum score of 55 points, based on the distribution of EP-like lesions in the cranial, middle, and accessory lobes, in addition to the apical edge of the caudal lobes.

Samples obtained for histopathological analysis included the interface between healthy and diseased lung tissue and were fixed by immersion in 10% neutral buffered formalin (NBF), embedded in paraffin wax, and routinely processed for histopathology. Section of 4 μm were stained using hematoxylin and eosin (H&E), and the microscopical lesions were scored using the protocol described by Livingston et al. [10] based on a semiquantitative histopathological scoring scheme ranging from 0 (negative) to 4 (severe). This method assesses the distribution and development of lymphoid nodules, the presence of inflammatory cells, and the development of perivascular and peribronchiolar lymphoid tissue, as well as the influence of these changes on the airways.

### 2.6. Laser Capture Microdissection

In addition to the samples processed for histopathological analysis, lung tissue samples were also snap frozen and embedded in OCT compound (Sakura Finetek, Newbury, UK), and 10 μm sections were cut using a cryostat as previously described [24]. Sections were placed in 2 μm polyethylene naphthalate membranes (Leica Microsystems, Wetzlar, Germany), air dried, and then fixed in 70% ethanol for 5 min, followed by staining with DNase-free H&E.

Tissue samples from bronchi, BALT, and lung parenchyma were collected in nuclease-free PCR tubes (Greiner Bio-One, Stonehouse, UK) using a Leica LMD6500 laser-capture microdissector (Leica Microsystems, Wetzlar, Germany). For this purpose, duplicate samples per each tissue target were collected. Each sample included five sections for each tissue type, which were subsequently processed to secure a representative concentration of RNA and DNA for downstream analysis. In order to reduce the variability in the samples used for Mhyo quantification, the same area of tissue was obtained from all tissues. Once dissected, samples were preserved at −80 °C until analysis.

### 2.7. DNA Extraction and Mhyo Quantification

QIAamp^®^ Micro Kit (Qiagen, Hilden, Germany) was used for isolation of bacterial DNA from the tissue samples. A specific qPCR targeting the p102 adhesin of Mhyo was used to quantify the bacterial load [30]. A plasmid control was prepared as positive control for standard curve preparation and qPCR DNA quantification was conducted using the pGEM^®^ Easy Vector System (Promega, Chilworth, UK) combined with One Shot™ TOP10 Chemically Competent *Escherichia coli* (Invitrogen, Paisley, UK). Plasmids were purified using the QIAprep^®^ Spin Miniprep kit (Qiagen, Hilden, UK) and confirmed by Sanger sequencing.

qPCR reactions were performed in a Stratagene MX3000P qPCR System (Stratagene, Milton Keynes, UK). Reactions were carried out following an initial denaturation at 95 °C for 2 min and 40 cycles of denaturation at 95 °C for 15 s and annealing ⁄ extension at 60 °C for 60 s.

### 2.8. RNA Extraction and qPCR Gene Expression Analysis

RNA was extracted using the RNAqueous™-Micro Total RNA Isolation Kit (Invitrogen, UK) following the manufacturer’s instructions. The total RNA obtained was then quantified using Qubit™ 2.0 Fluorometer (Invitrogen, UK), and RNA concentration was adjusted to 2 ng/μL for all samples.

Prior to the qPCR reactions, complementary DNA (cDNA) was obtained from the RNA samples using a cDNA synthesis kit (SuperScript™ VILO™, Invitrogen, UK), and the samples were stored at −80 °C until processing.

qPCR reactions were performed using a Stratagene MX3000P qPCR System (Stratagene, UK). EXPRESS qPCR Supermix (Invitrogen, UK) with ROX reference dye was used for the qPCR reactions. The reaction procedure was divided into an initial denaturation step at 95 °C for 2 min followed by 40 cycles of denaturation at 95 °C for 15 s and annealing ⁄ extension at 60 °C for 60 s. Gene targets, primer/probe sequences, and concentrations can be found in Table 1. The expression of cytokines was calculated using the 2^−ΔΔCt^ method [31]. RPL32 was selected as the reference gene for normalization due to its expression stability across porcine tissues [32]. Gene expression results are displayed as log_2_ fold change (FC).

### 2.9. Statistical Analysis

The one-sample Kolmogorov–Smirnov test was applied to test the normality of the distribution of variables. The Mann–Whitney U test and Kruskal–Wallis test with post hoc Bonferroni correction was used to test for differences between variables. The Spearman rank-order correlation coefficient was used for the correlation analysis between Mhyo quantification and gross pathology scoring, histopathology scoring, and ICAM1, IL-8, IL-10, IL-23, IFN-α, IFN-γ, TGF-β, and TNF-α expression. Statistical tests were performed using IBM^®^ SPSS^®^ Statistic version 26 (IBM Corp., Armonk, NY, USA). A *p*-value < 0.05 was considered significant. Figures were produced using GraphPad Prism^®^ version 10.2.2 (GraphPad Software, San Diego, CA, USA) and Microsoft Office^®^ 365 (Microsoft, Redmond, WA, USA), and collated using Adobe Photoshop^®^ CS4 (Adobe Systems Incorporated, San Jose, CA, USA).

## 3. Results

### 3.1. Mhyo Humoral Response Analysis

Blocking ELISA results can be found in Appendix A. Seropositivity was detected at 21 dpi in groups A, D, and E. At 28 dpi, seropositivity could be observed in groups A, C, D, and E.

Group E showed the highest humoral response against Mhyo infection, with 5/8 positive samples at 21 dpi, in addition to 2 suspects at the same sampling date, followed by a complete seropositivity (8/8 positive samples) at 28 dpi. Counts of 5/8 suspects and 3/8 positive animals were found at 21 dpi for group A, which progressed to 5/8 positives at 28 dpi. Group D revealed 2/8 positives at 21 dpi, progressing to 6/8 positives and 1/8 suspect samples at 28 dpi. Counts of 2/8 positive and 4/8 suspect animals were detected at 28 dpi for group C. All the other tested samples were found to be negative for the presence of specific Mhyo antibodies.

### 3.2. Gross Pathology and Histopathology

Gross pathology scores per animal ranged from 0 to 44 in the infected groups (Figure 1A). The highest mean scores were observed in group A (control strain 232, mean gross score: 23.6/55), followed by groups E (14/55), D (12/55), C (4.4/55), and B (1.9/55). A single animal in the negative control group presented a minimal score of 1/55 (mean gross score: 0.1/55). Statistically significant differences in gross pathology scores were observed between group A and groups B (*p*-value = 0.009), C (*p*-value = 0.046), and F (*p*-value < 0.001). In addition, groups D and E also presented statistically significant differences with the sham-infected group (*p*-value = 0.009 and 0.002, respectively).

Histopathology mean scores ranged from 0.9 to 3.6 in the infected groups (Figure 1B). Statistically significant differences were observed between group A and groups B (*p*-value = 0.008), C (*p*-value = 0.039), and F (*p*-value < 0.001). As described above, groups D and E also presented significant differences with the sham-infected group (*p*-value < 0.001 and 0.008, respectively). Furthermore, groups B and D presented significant differences in the histopathological score (*p*-value = 0.038).

### 3.3. Detection of Mhyo Nucleic Acids by qPCR

Mhyo DNA was not detected in any of the samples obtained from the control group. The highest Mhyo load in all groups was observed in bronchi samples, with an average of 1.4 × 10^3^ genome equivalents, followed by lung parenchyma (5 × 10^2^ genome equivalents) and BALT (10 genome equivalents). Significant differences between Mhyo DNA loads were found between bronchi and BALT qPCR results (*p*-value = 0.009).

Regarding each lung compartment, group D presented the highest concentration of Mhyo DNA in the bronchi (3.3 × 10^3^ genome equivalents), followed by group E (1.9 × 10^3^ genome equivalents) and group A (1.4 × 10^3^ genome equivalents) (Figure 2A). The concentration of Mhyo in BALT ranged from 19 genome equivalents for group D to 11 for group A (Figure 2B). Mhyo DNA detected in lung parenchyma was more abundant in group A (1.1 × 10^3^ genome equivalents), followed by group E (6.9 × 10^2^ genome equivalents) and group D (6.1 × 10^2^ genome equivalents) (Figure 2C).

Correlation results between Mhyo load in different lung compartments and gross and histopathologic score can be found in Table 2. A very strong correlation between gross pathology and histopathology scores was observed (*r*_s_ = 0.901; *p*-value ≤ 0.01). All the other correlation values were classified as strong, ranging from *r*_s_ = 0.791, for the correlation between histopathology score and Mhyo load in lung parenchyma, to *r*_s_ = 0.68, for the correlation between Mhyo load in bronchi and BALT.

### 3.4. Gene Expression

Differences in gene expression profiles per lung compartment were observed when all the infected animals were included in the analysis (Figure 3).

ICAM1 was slightly upregulated in the bronchi (log_2_ FC: 0.94), whereas BALT and lung parenchyma displayed a slight downregulation (log_2_ FC: −0.29 and −0.1, respectively). IL-8 was markedly upregulated in bronchi (log_2_ FC: 4.2) and lung parenchyma (log_2_ FC: −4.6), with a lesser response in BALT (log_2_ FC: 2.2). IL-10 also presented a relevant upregulation in bronchi (log_2_ FC: 3.4) and BALT (log_2_ FC: 3.1), while the expression levels were significantly lower in parenchyma (log_2_ FC: 0.9). IL-23 was slightly upregulated in all tissues, ranging from 1.6 to 1.8 log_2_ FC, with no significant differences between values observed. IFN-α downregulation was marked for BALT (log_2_ FC: −3.1). Additionally, a less marked downregulation was also observed for bronchi (log_2_ FC: −0.5) and parenchyma (log_2_ FC: −0.3). IFN-γ expression was relatively stable after Mhyo infection in bronchi and BALT, while downregulation of this cytokine was observed in lung parenchyma (log_2_ FC: −1.5). TGF-β upregulation was also mild, with log_2_ FC values ranging from 0.9 for BALT to 0.2 for parenchyma. A similar result was observed for TNF-α in bronchi and BALT, with a mild upregulation (log_2_ FC: 0.6 and 1.1, respectively), while a slight downregulation was detected in parenchyma (log_2_ FC: −0.6).

The influence of strain variability in the expression of immune mediators was also analyzed (Figure 4). The upregulation of IL-8 and IL-10 in BALT, and downregulation of IFN-α in the same lung compartment, in addition to the mild upregulation of IL-10 in lung parenchyma, were the only gene expression changes consistent with the five strains tested. Downregulation of TNF-α and IFN-α in lung parenchyma of animals infected with the Mhyo strains associated with highest tissue damage (groups A, D, and E) was observed, with no statistically significant differences between groups.

The correlation between gene expression results, bacterial load, and lung compartment demonstrated a significant strong correlation between Mhyo load and IL-8 gene expression in bronchi and parenchyma. A strong correlation was also observed between ICAM1 and IFN-α, and TNF-α expression in lung parenchyma, and with TGF-β in both bronchi and parenchyma. IL-8 expression in bronchi was strongly correlated with TGF-β expression. IL-10 and IFN-γ expression in lung parenchyma was also strong, as well as IL-23 with IFN-α expression in BALT. IFN-γ expression strongly correlated with TNF-α expression in the three lung compartments studied. In addition, a strong association between TGF-β expression in parenchyma and TNF-α was detected (Table 3).

## 4. Discussion

Mhyo is still considered one of the main concerns for the porcine industry worldwide due to its effects on animal health and the economic losses associated with EP. In this study, we examined the host–pathogen relationship between various Mhyo strains in experimentally infected piglets, with an emphasis on the pathological alterations in the lung tissue and the local immune response. Our results highlight the significance of strain variability in the development of macroscopic and microscopic lung lesions and the stimulation of the host’s immune response, shedding light on the complex dynamics in the host–pathogen interaction of EP.

ICAM1 upregulation is one of the initial factors in the adhesion of mycoplasmas to animal cells [34], favoring the migration of endothelial leukocytes and inducing a proinflammatory response. ICAM1 was used in our study to explore the suitability of LCM-derived samples to differentially express selected genes depending on the collected porcine lung tissues. Upregulation after Mhyo infection was clearly limited to the bronchi, linked to the direct stimulation of epithelial cells by Mhyo, as expected in cases of EP, due to initial pathogenic events associated with the adhesion to ciliated epithelial cells.

The activity of pro- and anti-inflammatory cytokines in cases of Mhyo infection has been previously linked to the pathogenesis of EP [13,14,17,35,36,37]. Our findings show that proinflammatory cytokine IL-8 was overexpressed in all the lung compartments analyzed, more significantly in bronchi and BALT, with minimal differences in expression among infection groups. This upregulation has been previously described in experimental infections with Mhyo, where IL-8 was the highest expressed cytokine in BALT, alveolar septa, and airway exudate, based on immunohistochemistry analysis [38]. Almeida et al. [39], in a study focused on evaluating the gene expression levels of different cytokines in the lung after Mhyo infection, also found IL-8 to be one of the most expressed cytokines, for up to 56 dpi, suggesting the relevance of this cytokine in the latter stages of EP. Other authors also found a statistically significant mRNA upregulation of IL-8 in tracheobronchial lymph nodes (TBLNs) from Mhyo-infected pigs [40], as well as in porcine alveolar macrophages (PAMs) [41]. These findings highlight the central role of IL-8 in EP. IL-8 production activates and attracts neutrophils, promoting the infiltration in lung parenchyma of pigs [42]. After neutrophil activation, monocytes, macrophages, and dendritic cell are subsequently activated, triggering the initiation of the adaptive immunity, thereby contributing to the lymphoproliferative response observed as part of Mhyo infections [40,43]. IL-8 expression has been positively correlated with histological scoring in EP cases [38], as well as with bacterial load [39]. In our study, a strong positive correlation between Mhyo load and the expression of IL-8 in bronchi and parenchyma, the two lung compartments with the highest bacterial counts based on qPCR, was observed. This correlation has been associated with the immunostimulant effect of lipoproteins, present on the surface of many mycoplasma species, which play a key role in the immunomodulation of Mhyo infection, activating the immune response by inducing the expression of inflammatory cytokines, such as IL-23 and TNF-α [43]. Interestingly, the reduction in lung lesions after Mhyo infections linked to vaccine protection has been linked to a reduction in IL-8 expression, probably associated with the decreased chemotactic effect of this chemokine [38]. This suggests that a potential modulation of the expression of IL-8 could have beneficial effects in controlling the tissue damage in EP.

IL-10 upregulation was also significant, with a relatively consistent expression among infected groups. The significant expression of IL-10 is a common immunomodulatory finding in cases of EP. Contrary to the results described for IL-8, no significant association between Mhyo load and IL-10 overexpression was observed, in accordance with previous reports [39]. Upregulation of this cytokine was more manifest in bronchi and BALT, with a mild expression in lung parenchyma. The expression of this cytokine in BALT is associated with the removal of Mhyo from the lung [44,45]. Additionally, alveolar macrophages and bronchoalveolar lavage fluid (BALF) have been found to be a source of IL-10 in cases of EP [35,37]. IL-10 mRNA expression analysis in dendritic cells demonstrated a strong upregulation of this cytokine [46], potentially aiding in the development of chronic respiratory disease [5]. On the contrary, some authors have a described a consistent IL-10 mRNA downregulation in the lung of Mhyo infected pigs for up to 56 dpi [39]. Interestingly, the same strain used in their experimental infection model was inoculated to piglets in group A of the present study, suggesting that those differences in gene expression could be associated with the potential induction of selective immunomodulation by Mhyo strains as a response to the host’s immune response.

IL-10, produced by activated macrophages and Th2-cells, is a potent anti-inflammatory cytokine, which reduces the expression of inflammatory cytokine such as TNF-α and alters the function of Th1 lymphocytes and macrophage function, potentially related to the delayed immune response against Mhyo [37,42]. For example, IL-10 overexpression has been described as a risk factor for the development of refractory *Mycoplasma pneumoniae* pneumonia in children [47], suggesting a central role for this cytokine in the development of chronic cases of respiratory infections caused by mycoplasma.

Apart from the potential role in the development of chronic respiratory disease, IL-10 also favors infection by other relevant pathogens involved in the PRDC, such as PCV2 and PRRSv, as IL-10 expression may be involved in the reduced clearance of the viruses [37,40], confirming the role of Mhyo as the primary pathogen in the PRDC and its function in synergistic co-infections.

Despite this association with the chronicity of EP and its synergistic effect with other pathogens in cases of the PRDC, IL-10 has been also linked to clinical protection and lung lesion reduction in bacterial infections [48]. For example, Beuckelaere et al. [49] demonstrated that reduced macroscopic lung lesions after Mhyo vaccination was associated with the suppression of pro-inflammatory cytokines and overexpression of IL-10. These data suggest that the interaction between Th1 and Th2 response is essential for the development of EP, and that control strategies based on increasing IL-10 expression versus Th1-biased immune response may be key in managing this disease.

IL-23 presented a moderate upregulation in the lung compartments analyzed, with minimal differences between infected groups, and no significant association with Mhyo load, although its expression has been linked to the stimulation of bacterial lipoproteins [43], as well IL-8. IL-23, secreted by dendritic cells and macrophages, belongs to the IL-12 cytokine family and is associated with the production of pro-inflammatory cytokines and the secretion of immunomodulatory cytokines such as TGF-β. In addition, its expression is required for the persistence of Th1 and Th17 cells [50]. While Th1 response in clearly associated with Mhyo infections, the data regarding the role of Th17 immune response in mycoplasma infections are limited, although evidence suggests that IL-23 is a relevant contributor to IL-17 expression, facilitating the recruitment of neutrophils in cases of mycoplasma pneumonia [51].

IFN-α was negatively regulated in 4 out of 5 groups, with a weak upregulation in bronchi and parenchyma of group B, the experimental group with the lowest gross pathology and histopathology scores, indicating a potential correlation between the development of lung lesions and the downregulation of this cytokine. Consistent downregulation of IFN-α in the three lung compartments was associated with those groups, with a higher score in macroscopic and microscopic lung lesions. In regard to Mhyo load, negative correlations were observed for all compartments, with significant results observed in lung parenchyma. IFN-α is a highly relevant cytokine in the initial immunological reaction against viral infection [52]; however, its role in mycoplasma infection has not been completely elucidated. In regard to EP, the downregulation of IFN-α has been previously described in different cells after Mhyo stimulation. Zhang et al. [40] described a mild mRNA downregulation for this cytokine in TBLN after Mhyo infection. This downregulation became strongly evident when a co-infection with PCV2 occurred. This virus is able to induce inhibitory effects in the production of IFN-α on its own [53]; however, the synergistic effect found in co-infections with Mhyo suggests that this cytokine is also a relevant actor in the immunomodulatory repertoire of Mhyo. This synergistic effect with other members of the PRDC, such as PRRSv, has also been described. Li et al. [54] observed that a secretory nuclease previously described as a virulence factor for Mhyo (Mhp597) was able to suppress the expression of IFN-α in PAMs while promoting the replication of PRRSv. The present study demonstrated that IFN-α downregulation after Mhyo infection in bronchi, BALT, and lung parenchyma is a relevant factor in the immunological response against EP and its role in the development of the PRDC should be further characterized.

IFN-γ mRNA expression varied between lung compartments and experimental groups, with mild upregulation in bronchi and BALT combined with an evident downregulation in lung parenchyma. Group E displayed a consistent downregulation for this cytokine in all lung compartments. Similar results were observed in the lungs of Mhyo-infected pigs, where a mild upregulation of IFN-γ mRNA was found in the early stages of the disease, followed by a downregulation 8 weeks after the infection [39]. Stimulation of dendritic cells from the nasal cavity of pigs was also associated with a decrease in IFN-γ mRNA expression [46]. IFN-γ is a relevant neutrophil and macrophage activator, as well as playing a relevant role in the differentiation of Th1 cells, which is a known efficient response to mycoplasma infections [42,46]. In addition, IFN-γ can produce a direct antiviral action on infected cells [55]. Therefore, the suppressed expression of this cytokine may also play a role in perpetuating EP by reducing the potential resistance effect of Th1 response versus Th2 and favoring the development of co-infections with viruses involved in the PRDC after Mhyo infection. Interestingly, increased levels of IFN-γ in vaccinated pigs versus infected pigs have been described, demonstrating the potential effect of IFN-γ overexpression in the protection against Mhyo-vaccinated animals [38]

TNF-α is another cytokine frequently associated with Mhyo infections [17,35,36]. TNF-α production is related to the stimuli produced by mycoplasmal lipoprotein in activated macrophages and monocytes [43]. Contrary to other findings [39], no correlation with bacterial load and TNF-α overexpression could be detected in the present study. This result can be explained by the fact that the production of this cytokine is linked to the initial inflammatory response in cases of EP, with peak overexpression responses between 7 and 21 dpi [36,39], decreasing after 28–35 dpi. Consequently, the date of sample collection in our study (28 dpi) may be related to the lack of a significant elevation in the expression levels of this cytokine.

TGF-β, an immunosuppressive cytokine involved in macrophage and Th1 activity inhibition [42], was evaluated in order to understand the role of other immunosuppressive mediators in Mhyo infection. TGF-β expression between lung compartments and experimental groups after Mhyo infection was highly variable. TGF-β has been used as a valuable biomarker in cases of *Mycoplasma bovis* pneumoniae in calves due to its association with inflammation and damage of the lung [56]; however, based on our results, further research should be conducted to elucidate the role of TGF-β as an immunomodulatory factor in Mhyo infections.

Although our study provides similar results in comparison to previous in vivo studies assessing Mhyo infection and host–pathogen interaction, and diagnostic tests, continuous health evaluation, and strict biosecurity strategies were put in place to avoid contamination by other porcine pathogens, the potential effect of not using SPF pigs should be taken into consideration as a limitation of the present study when analyzing the results.

In summary, our study delved into the complex relationships among lung lesions, immune response in different lung compartments, and strain diversity in regard to the host–pathogen interaction in piglets experimentally infected with Mhyo. This study stressed the need for an in-depth understanding of EP pathogenesis in order to allow the tailoring of specific control measures and improved vaccines that could benefit porcine health worldwide.

## 5. Conclusions

The results presented in this study confirm the applicability of LCM for the analysis of the host–pathogen interaction in cases of EP. Gene expression analysis of porcine lung compartments demonstrated the differential up- and downregulation of multiple cytokines involved in the immune response against Mhyo infection. The interplay between Th1 and Th2 immune response was proven to be central in the infection caused by this mycoplasma, where IL-8 and IL-10 stood out as the main cytokines involved in EP. Control strategies based on the modulation of the expression of these cytokines could be relevant to the prophylaxis of EP. In addition, the upregulation of IL-23 suggests the potential role of Th17 response in Mhyo infections, although further research should be performed in order to confirm this assumption. The marked downregulation of IFN-α suggests a potential survival response of Mhyo to the host’s immune response, aimed at increasing tissue damage by collaborating synergistically with viruses involved in the PRDC. Due to the complexity of host–pathogen interaction in cases of EP, future studies should be aimed at building on our knowledge regarding the transcriptional changes in the lungs of infected pigs after Mhyo infection so better prophylactic control strategies against EP can be developed.

## Figures and Tables

**Figure 1 animals-14-01290-f001:**
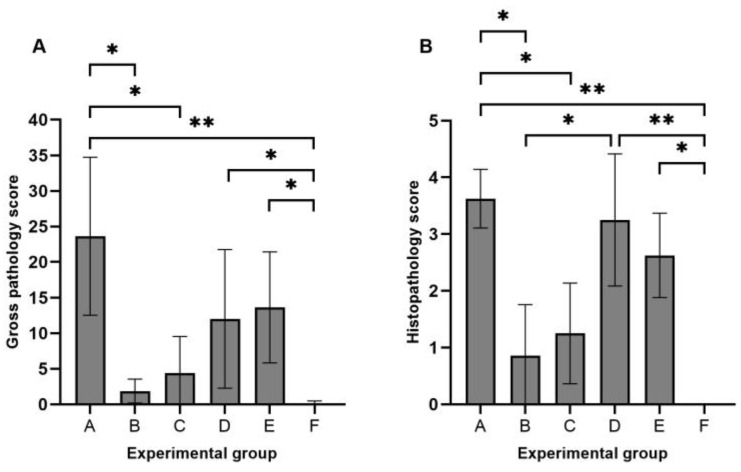
Gross pathology (**A**) and histopathology scoring (**B**). * *p*-value < 0.05; ** *p*-value < 0.001.

**Figure 2 animals-14-01290-f002:**
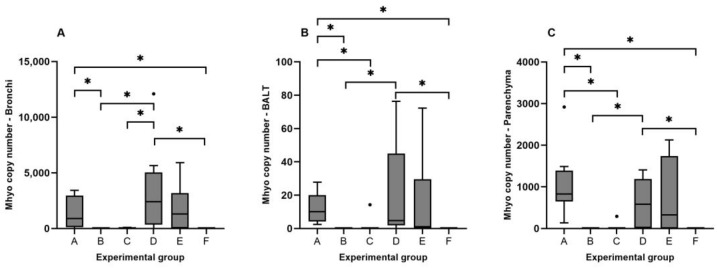
Box plots of Mhyo DNA quantification results in bronchi (**A**), BALT (**B**), and lung parenchyma (**C**). * *p*-value < 0.05.

**Figure 3 animals-14-01290-f003:**
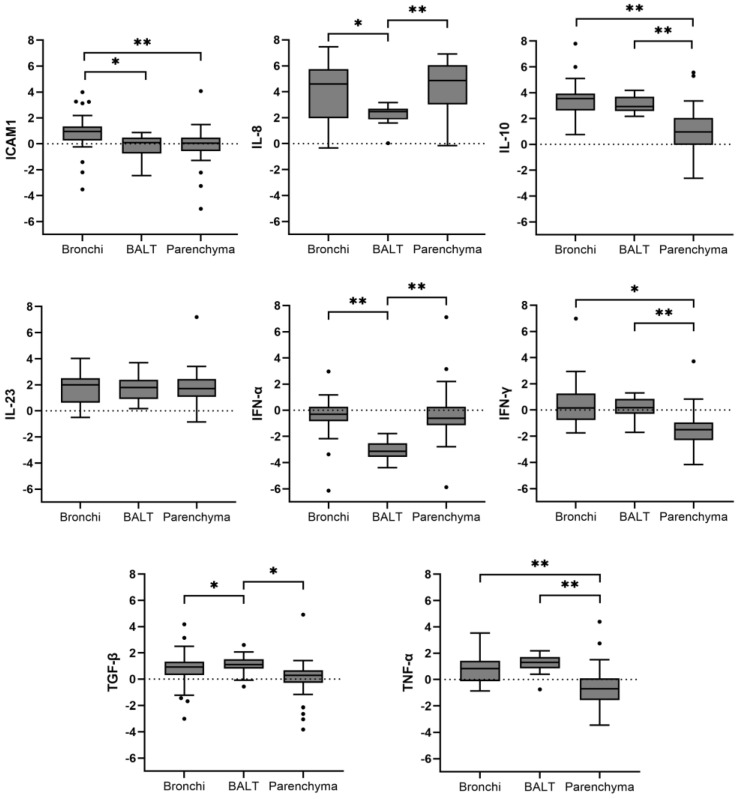
Differential gene expression per lung compartment for each gene tested. The y axis represents log_2_ fold change of gene expression *: *p*-value < 0.05; **: *p*-value < 0.001.

**Figure 4 animals-14-01290-f004:**
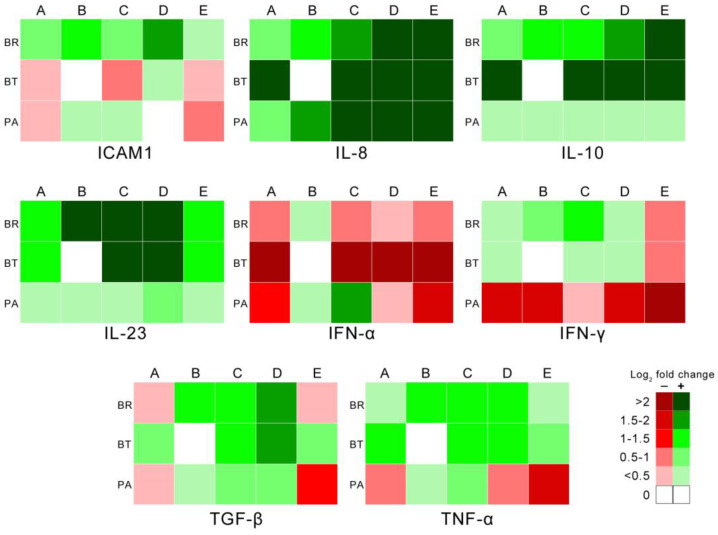
Strain influence in gene expression per lung compartment for each gene tested. Animal groups are identified from A to E. BR: bronchi; BT: BALT; PA: lung parenchyma.

**Table 1 animals-14-01290-t001:** Primers and probes used in the gene expression analysis.

Gene	Forward Primer (5′-3′)	Reverse Primer (5′-3′)	Probe (5′-3′)	Reference
ICAM1	GGACCATGGGCGAAGCTT	TGGGCAATCCCTCTCGTCTA	[6FAM]-ACATGCCCAGCCACCGTCAGG-[TAM]	[33]
IL-8	CCGTGTCAACATGACTTCCAA	GCCTCACAGAGAGCTGCAGAA	[6FAM]-CTGTTGCCTTCTTGGCAGTTTTCCTGC-[TAM]	[33]
IL-10	TGAGAACAGCTGCATCCACTTC	TCTGGTCCTTCGTTTGAAAGAAA	[6FAM]-CAACCAGCCTGCCCCACATGC-[TAM]	[33]
IL-23	AGAAGAGGGAGATGATGAGAC	AGCAGGACTGACTGCCGTCC	[6FAM]-CTGAGGATCACAGCCATCCCCGC-[TAM]	[24]
IFN-α	TCAGCTGCAATGCCATCTG	AGGGAGAGATTCTCCTCATTTGTG	[6FAM]-TGACCTGCCTCAGACCCACAGCC-[TAM]	[33]
IFN-γ	GAAAAGCTGATTAAATTCCGGTAG	AGGTTAGATCTTGGTGACAGATC	[6FAM]-TCTGCAGATCCAGCGCAAAGCCATCAG-[TAM]	[24]
TGF-β	AGGGCTACCATGCCAATTT	CCGGGTTGTGCTGGTTGT	[6FAM]-CACTCAGTACAGCAAGGTCCTGGCTCTGTA-[TAM]	[33]
TNF-α	TGGCCCCTTGAGCATCA	CGGGCTTATCTGAGGTTTGAGA	[6FAM]-CCCTCTGGCCCAAGGACTCAGATCA-[TAM]	[33]
Rpl32	TGGAAGAGACGTTGTGAGCAA	CGGAAGTTTCTGGTACACAATGTAA	[6FAM]-ATTTGTTGCACATTAGCAGCACTTCAAGCTC-[TAM]	[33]

TGF: Transforming growth factor. ICAM1: intercellular adhesion molecule 1. FAM: fluorescein. TAM: Tamra.

**Table 2 animals-14-01290-t002:** Spearman’s correlation matrix for gross pathology score, histopathology score, and Mhyo quantification by lung compartment.

	Gross Pathology Score	Histopathology Score	Mhyo Load—Bronchi	Mhyo Load—BALT	Mhyo Load—Parenchyma
Gross pathology score		0.901	0.699	0.733	0.75
Histopathology score	0.901		0.769	0.753	0.791
Mhyo load—bronchi	0.699	0.769		0.68	0.732
Mhyo load- BALT	0.733	0.753	0.68		0.879
Mhyo load—parenchyma	0.75	0.791	0.732	0.879	

All correlation values are statistically significant (*p*-value ≤ 0.01).

**Table 3 animals-14-01290-t003:** Spearman’s correlation matrix for Mhyo quantification and ICAM1, IL-8, IL-10, IL-23, IFN-α, IFN-γ, TGF-β, and TNF-α by lung compartment.

		ICAM1	IL-8	IL-10	IL-23	IFN-α	IFN-γ	TGF-β	TNF-α
Mhyo load	BR	0.147	0.663 **	0.194	−0.215	−0.294	−0.279	−0.028	−0.373 *
	BT	0.065	0.329	0.037	−0.156	−0.05	0.381	0.273	−0.077
	PA	−0.323	0.663 **	0.099	−0.483 **	−0.571 **	−0.022	−0.131	−0.241
ICAM1	BR		0.380 *	0.514 **	0.350 *	0.285	0.399 *	0.741 **	0.559 **
	BT		0.028	−0.259	0.097	0.534 *	0.206	0.578 *	0.544 *
	PA		0.018	0.2	0.497 **	0.633 **	0.496 **	0.647 **	0.726 **
IL-8	BR			0.514 **	0.350 *	0.285	0.399 *	0.741 **	0.559 *
	BT			−0.383	−0.318	0.058	−0.258	0.061	−0.548
	PA			0.192	−0.059	−0.268	−0.008	0.054	−0.06
IL-10	BR				0.374	−0.071	0.165	0.412 *	0.269
	BT				0.168	−0.274	−0.059	0.009	−0.106
	PA				0.202	−0.065	0.610 **	0.371 *	0.463 **
IL-23	BR					0.362 *	0.15 *	0.405 *	0.706 **
	BT					0.606 *	0.041	0.512 *	0.021
	PA					0.512 **	0.294	0.540 **	0.462 *
IFN-α	BR						0.314	0.392 *	0.497 **
	BT						−0.015	0.446	0.243
	PA						0.169	0.474 **	0.485 **
IFN-γ	BR							0.519 **	0.735 **
	BT							0.211	0.608 **
	PA							0.517 **	0.783 **
TGF-β	BR								0.573 **
	BT								0.318
	PA								0.708 **

* *p*-value ≤ 0.05. ** *p*-value ≤ 0.01. BR: bronchi; BT: BALT; PA: lung parenchyma.

## Data Availability

The data presented in this study can be obtained upon request from the corresponding author.

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
