# Peer review of "Differential Gene Expression in Porcine Lung Compartments after Experimental Infection with Mycoplasma hyopneumoniae"

_animals, 2024, doi:10.3390/ani14091290_

Round 1
Reviewer 1 Report
Comments and Suggestions for Authors
The manuscript entitled reports the findings of cytokine expression in lung compartments of Mhyo experimentally challenged pigs. Overall, the manuscript is very well written and brings some uniqueness, such as the use of several Mhyo strain to experimentally challenge pigs – which is considered challenging in comparison to other less fastidious bacteria - and the laser microdissection, a technique that requires time and training to be properly performed.
However, I have a main concern that need to be addressed before the manuscript is finally considered for publication. The concern is about ruling out possible co-infections that could directly influence on the results. If samples (serum, nasal swabs, and lung tissue) were not screened for other potential co-infections I strongly recommend the authors go back to archived samples and screen them for the most common pathogens that can colonize and infect lungs, such as Influenza A, PCV2, PRRS, Mycoplasma hyorhinis, Pasteurella multocida, Glaesserella parasuis, and others. If such screening was performed, provide results as supplementary and add a sentence in Results stating that the tests were performed, followed by a summary of the results.
Simple summary: Overall, the simple summary is difficult to follow, since the objectives were not clearly stated and there are results that do not correspond to the methodologies presented. Example: Why and how Mhyo strain virulence was evaluated?
Page 1, lines 25 and 26. It is not clear why the understanding of the interaction between Mhyo and the porcine host is crucial. Please, explain that better.
Page 1, line 29: For a simple summary it would be more beneficial to remove the cytokine’s names and better justify the relevance of the research.
Page 1, lines 28-30. The way the sentence is written may confuse the reader, as macro and microscopic evaluation are not related to the laser-capture microdissection. Consider rephrasing and inverting the order of the methodologies in the sentence.
Page 1, line 41: The relevance of the understanding the interaction between Mhyo and the porcine host was not stated. Please, provide at least one reason why this knowledge is relevant.
Page 2, lines 53 and 54: What could be the application of the results obtained by this study?
Page 2, lines 71 and 72: Colonization by what? Mhyo? Other secondary/opportunistic bacteria? Please, clarify.
Page 3, last paragraph of introduction: Although well written, the last paragraph is not truly necessary for the understanding of the research. Consider removing it to make the introduction more concise.
Page 3, lines 162 and 163: I believe you meant that the inoculation was performed after isoflurane anesthesia. If that is correct, please, fix the sentence.
Page 4, lines 169 and 170: Remove the repeated words (was performed).
Page 7, table 1: In the bottom of the page, it is mentioned that one pig from group B had to be euthanized due to welfare issues. Animal welfare is a very broad concept and thus, the reasons/clinical signs presented by the pig need to be clearly stated. If a necropsy or any laboratorial tests were performed on the euthanized pig to determine etiology of the condition, provide the information as well.
Page 9, 3.3 Gene expression: Results obtained for each group should be included as supplementary materials.
Page 10, figure 4: Image quality needs to be increased.
Page 11, table 3: This table should be presented in one page.
Page 12, table 365: Replace animal welfare with animal health or be more specific why Mhyo influences in animal welfare, since the concept of animal welfare is quite broad.
Comments on the Quality of English LanguageMinor changes need to be addressed as listed above.
Author Response
REVIEWER 1.
The manuscript entitled reports the findings of cytokine expression in lung compartments of Mhyo experimentally challenged pigs. Overall, the manuscript is very well written and brings some uniqueness, such as the use of several Mhyo strain to experimentally challenge pigs – which is considered challenging in comparison to other less fastidious bacteria - and the laser microdissection, a technique that requires time and training to be properly performed.
However, I have a main concern that need to be addressed before the manuscript is finally considered for publication. The concern is about ruling out possible co-infections that could directly influence on the results. If samples (serum, nasal swabs, and lung tissue) were not screened for other potential co-infections I strongly recommend the authors go back to archived samples and screen them for the most common pathogens that can colonize and infect lungs, such as Influenza A, PCV2, PRRS, Mycoplasma hyorhinis, Pasteurella multocida, Glaesserella parasuis, and others. If such screening was performed, provide results as supplementary and add a sentence in Results stating that the tests were performed, followed by a summary of the results.
We completely agree with the comments. All animals were tested on arrival for general bacteriological and virological analysis, however, in the manuscript we only focussed on the test applied to mycoplasma detection. The manuscript has been amended accordingly and also a paragraph regarding the limitations of the study has been included.
Simple summary: Overall, the simple summary is difficult to follow, since the objectives were not clearly stated and there are results that do not correspond to the methodologies presented. Example: Why and how Mhyo strain virulence was evaluated?
Thanks for the comment. The simple summary has been modified.
Page 1, lines 25 and 26. It is not clear why the understanding of the interaction between Mhyo and the porcine host is crucial. Please, explain that better.
Page 1, line 29: For a simple summary it would be more beneficial to remove the cytokine’s names and better justify the relevance of the research.
Page 1, lines 28-30. The way the sentence is written may confuse the reader, as macro and microscopic evaluation are not related to the laser-capture microdissection. Consider rephrasing and inverting the order of the methodologies in the sentence.
Thanks for the comments. We hope these comments are properly addressed with the changes performed in the simple summary.
Page 1, line 41: The relevance of the understanding the interaction between Mhyo and the porcine host was not stated. Please, provide at least one reason why this knowledge is relevant.
Thanks for the comment. A sentence has been added to highlight this relevance.
Page 2, lines 53 and 54: What could be the application of the results obtained by this study?
Page 2, lines 71 and 72: Colonization by what? Mhyo? Other secondary/opportunistic bacteria? Please, clarify.
The text has been modified to account for this comment
Page 3, last paragraph of introduction: Although well written, the last paragraph is not truly necessary for the understanding of the research. Consider removing it to make the introduction more concise.
We consider that this paragraph condenses the objectives of our study and therefore it is needed to lead the reader to the following sections of the manuscript.
Page 3, lines 162 and 163: I believe you meant that the inoculation was performed after isoflurane anesthesia. If that is correct, please, fix the sentence.
Thanks for the comment, corrected as suggested.
Page 4, lines 169 and 170: Remove the repeated words (was performed).
Deleted as suggested. Thanks for the comment
Page 7, table 1: In the bottom of the page, it is mentioned that one pig from group B had to be euthanized due to welfare issues. Animal welfare is a very broad concept and thus, the reasons/clinical signs presented by the pig need to be clearly stated. If a necropsy or any laboratorial tests were performed on the euthanized pig to determine etiology of the condition, provide the information as well.
Thanks for the comment. We meant welfare issues based on the humane endpoint scoring sheet. The clinical scoring protocol associated to the humane endpoints was performed by the designated veterinarian, recording non-specific clinical signs, including anorexia and respiratory distress for this pig prior sacrifice. Post-mortem examination did not show any specific lesions while Mhyo was the only pathogen detected in the respiratory tract, although no EP-like lesions were observed. This information has also been added to the manuscript
Page 9, 3.3 Gene expression: Results obtained for each group should be included as supplementary materials.
Results per group can be found in figure 4 (stated as A, B, C, D, E). As no statistical differences between groups were found, a heat map was used to better display the results, as we consider this type of graphic as a more visually pleasant strategy to analyse gene expression changes.
Page 10, figure 4: Image quality needs to be increased.
All figures have been modified in order to improve their quality.
Page 11, table 3: This table should be presented in one page.
Done.
Page 12, table 365: Replace animal welfare with animal health or be more specific why Mhyo influences in animal welfare, since the concept of animal welfare is quite broad.
Welfare was modified for health
Reviewer 2 Report
Comments and Suggestions for Authors
The work is prepared at a very good level and I recommend it for publication after slight modifications. In the material and methodology section, more information regarding the experimental animals used needs to be added. It is necessary to state the age and weight of the animals used and when they were weaned, as well as the gender composition of the piglets. Also state how the piglets were randomized into the experimental groups. Next, state what kind of feed was used, that is, the name and manufacturer. Please fill in the decision number of the Ethics Committee for the Protection of Animals against Cruelty approving this experiment.
Author Response
REVIEWER 2:
The work is prepared at a very good level and I recommend it for publication after slight modifications. In the material and methodology section, more information regarding the experimental animals used needs to be added. It is necessary to state the age and weight of the animals used and when they were weaned, as well as the gender composition of the piglets. Also state how the piglets were randomized into the experimental groups. Next, state what kind of feed was used, that is, the name and manufacturer. Please fill in the decision number of the Ethics Committee for the Protection of Animals against Cruelty approving this experiment.
Thanks for the comment. Age is displayed in the section 2.2. (5-week-old). Gender composition has been added. Weight has been added. Pig grouping is described in section 2.2 as well as the ethic committee approval number. All animals were weaned before transportation (4-week-old). A commercial feed was provided to all animals.
Reviewer 3 Report
Comments and Suggestions for Authors
It is an interesting study with important findings. However, the manuscript still needs more work to be published. I am describing below the improvements that, in my opinion, are essential to make the manuscript clearer, better illustrated and scientifically interesting:
1) It is necessary to rewrite the Simple Summary, since it is almost a repetition of the Abstract. Please highlight the importance of the study instead of describing the methods and the main results (they are already in the Abstract!). The first two paragraphs of the Introduction have some important information to be included in the Simple Summary.
2) The authors write in the Conclusion: “The results presented in this study confirms the usefulness of LCM for the analysis of the host-pathogen interaction in cases of EP.” However, the results do not describe the usefulness of LCM! Please, rewrite the results to demonstrate how this methodology (LCM) was important to analyze HP interaction. It is also necessary to discuss this aspect.
3) It is a little bit confuse the description of the objectives (lines 124-133). Please rewrite them! Remember these objectives must be directly related to the title of the manuscript!
4) There are unnecessary tables and figures, as for example Table 1 and Figure 1. They are taking the focus away from important results! Tip: transfer to supplementary material. I also recommend that authors review the other tables and figures in order to present the results in a better elaborated and informative way.
5) It is not possible to definitively establish the relationship between MH infection and the production of interleukins and interferons based on this study. Remember that the situation is more complex. Just one example: how can the authors be sure that the animals did not have other infections? Therefore, I recommend including a paragraph about the limitations of the study.
Finally, I recommend the authors to review carefully the complete manuscript before the resubmission and the next per review.
Comments on the Quality of English LanguageOk!
Author Response
REVIEWER 3:
It is an interesting study with important findings. However, the manuscript still needs more work to be published. I am describing below the improvements that, in my opinion, are essential to make the manuscript clearer, better illustrated and scientifically interesting:
- It is necessary to rewrite the Simple Summary, since it is almost a repetition of the Abstract. Please highlight the importance of the study instead of describing the methods and the main results (they are already in the Abstract!). The first two paragraphs of the Introduction have some important information to be included in the Simple Summary.
Thanks for the comment. The simple summary has been modified.
- The authors write in the Conclusion: “The results presented in this study confirms the usefulness of LCM for the analysis of the host-pathogen interaction in cases of EP.” However, the results do not describe the usefulness of LCM! Please, rewrite the results to demonstrate how this methodology (LCM) was important to analyze HP interaction. It is also necessary to discuss this aspect.
Thanks for the comment, the conclusion has been modified, in order to avoid any misunderstanding with the word usefulness.
- It is a little bit confuse the description of the objectives (lines 124-133). Please rewrite them! Remember these objectives must be directly related to the title of the manuscript!
Thanks for the comment, modified as suggested.
- There are unnecessary tables and figures, as for example Table 1 and Figure 1. They are taking the focus away from important results! Tip: transfer to supplementary material. I also recommend that authors review the other tables and figures in order to present the results in a better elaborated and informative way.
Thanks for the comments. The authors previously discussed the usefulness of the presence of figure 1 in the text, however, based on the comments from the reviewer, we will move it to the supplementary material as suggested. Regarding the rest of the figures and tables we have modified the figures in order to improve their quality.
- It is not possible to definitively establish the relationship between MH infection and the production of interleukins and interferons based on this study. Remember that the situation is more complex. Just one example: how can the authors be sure that the animals did not have other infections? Therefore, I recommend including a paragraph about the limitations of the study.
Thanks for the comment. A better description of the diagnostic tests performed to account for potential contamination/carrier status carried in the pigs used for the study has been added to the text. In addition a paragraph about the limitation of the study has been added to the discussion.
Finally, I recommend the authors to review carefully the complete manuscript before the resubmission and the next per review.
Round 2
Reviewer 1 Report
Comments and Suggestions for Authors
The authors have adequately addressed the concerns and made the necessary changes to improve the manuscript, which is now acceptable for publication.
Author Response
Many thanks for your constructive review
Reviewer 3 Report
Comments and Suggestions for Authors
The authors revised the manuscript according to some of my recommendations. The text is clearer and the article is better organized. However, there is still a need for better elaboration of the Simple Summary text and a more appropriate sentence about the objectives of the study at the end of the Introduction.
Comments on the Quality of English LanguageMinor (but necessary!) editing of English language required
Author Response
Dear reviewer, many thanks for your constructive review. Following your recomendation, in this new version of the manuscript we have modified the simple summary.